# Therapeutic Applications and Effects of *Lupinus angustifolius* (Blue Lupin) and Its Components: A Systematic Review and Meta-Analysis

**DOI:** 10.3390/foods12142749

**Published:** 2023-07-19

**Authors:** Rafael Fernández Castillo, Raquel García Pérez, Ana González Díaz, Antonio Liñán González

**Affiliations:** 1Faculty of Health Sciences, University of Granada, Parque Tecnológico de Ciencias de la Salud, Avd. de la Ilustración, 60, 18016 Granada, Spain; raquelgp@ugr.es (R.G.P.); antoniolg@ugr.es (A.L.G.); 2Faculty of Health Sciences, University of Granada, C/Cortadura del Valle s/n, 51001 Ceuta, Spain; agonzalezd@ugr.es

**Keywords:** lupin protein, plasma lipids, hypercholesterolemic subjects, human study, type 2 diabetes mellitus, glycemic control

## Abstract

*Lupinus angustifolius* has a unique nutrient profile among legumes and may have beneficial health effects when included in the diet. The aim of this study was to investigate the biological properties of *blue lupin (Lupinus angustifolius*), its chemical components, and their relevance for monitoring biological and anthropometric health markers, including triglycerides (TGs), low-density lipoprotein cholesterol (LDL-C), BMI, weight, and glycemia, compared with control groups with other kinds of diets. PubMed, Web of Science, and Scopus databases, updated to December 2023, were searched. Out of the 194 studies identified, a total of 7 randomized controlled trials (RCTs) comprising 302 participants met the eligibility criteria. The results of our study indicated that the blue lupin diet has a direct relationship with parameters such as blood glucose, weight, and LDL-C but not with TGs or BMI. In conclusion, the research described in this review clearly indicates that *L. angustifolius* may play an important role in the dietary prevention of hyperlipidemia and hypertension. Therefore, it would be highly advisable to increase its consumption in diets. However, further studies, ideally in humans, are required to truly establish *L. angustifolius*’s health-promoting properties.

## 1. Introduction 

Currently, there is growing demand for the use of plant proteins as functional food components in the nutraceutical industry and as alternatives to the environmentally costly production of animal proteins [1]. In this respect, legumes represent a dietary and economical source of good-quality protein, possessing a higher protein content than the majority of animal proteins and plant foods [2]. On the other hand, legumes, particularly lupin proteins and lupin peptides, provide beneficial health effects [3]. In this line, the blue lupin, or *Lupinus angustifolius*, is an annual herbaceous plant of the legume family, one of the few cultivated species of the genus *lupinus*, whose fruit is widely used in foods, e.g., as part of a typical snack of the Mediterranean region [4,5]. It stands out mainly for its high nutritional content, in which the protein and fiber contents represent up to 40% and 24%, respectively, of its dry weight. Additionally, its carbohydrate content represents less than 24% of its dry weight. Its high protein content makes it a suitable vegetable protein alternative to meat and soybeans [6]. Due to its remineralizing properties, highlighting its iron (7.6 mg) and calcium (180 mg) contents, the blue lupin also provides zinc, potassium, phosphorus, magnesium, B vitamins, and vitamin E [7,8,9]. Although we see that the contribution of lupine in fats is high, we must take into account that we are talking about fatty acids, whose contribution is beneficial [10]. The lupin diet provides important benefits, which include the following: the stimulation of cell renewal; the regulation of blood glucose levels; a reduction in uric acid and cholesterol levels; and the stimulation of intestinal transit due to its fiber and trace element contents [11,12,13]. According to scientific evidence, conglutin proteins from the leguminous plant blue lupin have been proven to have anti-inflammatory and antioxidant activities [8,14,15]. Therefore, natural products with nutraceutical properties may represent a promising therapeutic strategy for the prevention and treatment of diseases [16]. Based on their multiple beneficial effects on the amelioration of inflammatory-based diseases at the molecular level, conglutin proteins can be employed in the prevention and treatment of inflammation-related diseases, including obesity, diabetes [17,18], and cancer [19]. The phytochemicals present in *Lupinus* have been shown to play a key role in preventing various chronic-inflammation-related diseases such as diabetes or hypertension because of the antioxidant, antihyperlipidemic, and anti-inflammatory activities they possess [20]. Regarding phytochemicals, *Lupinus* is rich in polyphenols, phytosterols, and squalene. The most abundant phenolic compounds present in *L. angustifolius* seeds are flavones, phenolic acids, and isoflavones, which represent 76%, 19%, and 4% of the total identified phenols, respectively [21]. The aim of this study was to investigate the biological properties of blue lupin (*Lupinus angustifolius*), its chemical components, and their relevance for monitoring biological and anthropometric health markers.

## 2. Materials and Methods

This review and meta-analysis were carried out based on the guidelines proposed by PRISMA22 (Preferred Reporting Items for Systematic Reviews and Meta-Analyses). We describe the methodology used to systematically present our findings on the beneficial effects of lupin on chronic diseases. The protocol for this systematic review was registered in protocolos.io (DOI: https://dx.doi.org/10.17504/protocols.io.e6nvwjw6wlmk/v1 accessed on 14 May 2023).

### 2.1. Search Strategy 

We systematically searched online medical databases, including Web of Science, PubMed/Medline, and SCOPUS, for entries up to December 2022. The following search terms were used: (“*lupinus* diet” OR “*Lupinus angustifolius* dieting” OR “randomized controlled trial” OR “low carbohydrate lupinus diet” OR “very low carbohydrate *lupinus* diet”) AND (“body composition” OR “fat-free mass” OR “lean body mass” OR “LBM” OR “FFM”) AND (“resistance insulin” OR “muscle” OR “muscle mass” OR “diabetes” OR “blood lipids” OR “antioxidants” OR “cancer” OR “anti-inflammatory”). 

### 2.2. Study Selection 

Titles and abstracts of all articles obtained from the initial search were individually reviewed by the author. Studies published in Spanish and English and in the last 10 years were selected in order to review the most current evidence; in this regard, randomized trials with a minimum duration of two weeks and analytical studies that evaluated the effects of *Lupines angustifolius* therapy and its beneficial effects on chronic diseases were included. Studies that were not related to the subject or that did not provide relevant statistical information, research conducted on animals, systematic reviews or meta-analyses or uncontrolled experimental studies, and research without a control group were excluded. 

### 2.3. Data Extraction 

From each selected article information was extracted, including the name of the first author, the date of publication, the average age of the participants, and the gender of both the experimental and control groups, in addition to the study design, the participants, the duration, the composition of the diet, and the means and standard deviations (SD) of both groups. Data related to body weight, fasting glucose, BMI, LDL-C, and TGs were selected. 

### 2.4. Quality Assessment 

A risk of bias assessment was conducted using the Cochrane method [22]. The elements evaluated were the following: random sequence generation, allocation concealment, blinding of participants and personnel, blinding of outcome assessment, incomplete outcome data, selective reporting, and other bias. 

Studies were classified as high risk of bias, low risk of bias, or unclear bias for each item assessed based on the recommendations of the *Cochrane Handbook*. 

RevMan Web software (London, UK) was used to develop meta-analyses. Meta-analyses were performed assessing the difference in post-intervention means to estimate the heterogeneity of the included studies. The inconsistency statistic (I) was used, understanding heterogeneity as low if I^2^ < 50%, moderate if I^2^ 50–75%, and high if I^2^ > 75%, with a meta-analysis being recommended when the I2 was low or moderate. Publication bias was assessed using funnel plots.

## 3. Results

The flow of studies in our meta-analysis is depicted in Figure 1, from 194 possibly relevant references. References were reviewed to determine eligibility. Finally, only seven investigations met the eligibility criteria. These studies included 152 participants consuming a *Lupinus angustifolius* diet and 150 participants in control groups.

The characteristics of these five randomized controlled trials are presented in Table 1.

From the analyzed studies, six presented the effects of the lupin diet on participants’ body weight [23,24,26,27,28,29], LDL-C, and TGs [23,24,25,26,27,28,29]. Five studies showed additional effects on body mass index [23,24,26,27,28], and two studies described its effects on glycemia [28,29]. None of the analyzed variables showed the existence of publication bias.

### 3.1. Effects on Blood Glucose

A pooled data analysis consisting of 72 subjects and two studies showed that the *Lupinus angustifolius* diet had a positive impact on blood glucose levels. The mean difference and corresponding 95% confidence interval were −7.16 (−11.47, −2.84) in favor of the experimental group, and the differences were considered significant (*p* = 0.0001) (Figure 2).

### 3.2. Effects on Weight

For the weight variable, six studies provided data from 236 subjects which showed that the *Lupinus angustifolius* diet had a positive impact on weight. The mean difference and corresponding 95% confidence interval were −2.03 (−4.27, 0.20) in favor of the experimental group, although the differences were not considered significant (*p* = 0.07) (Figure 3).

### 3.3. Effects on LDL-C

For the LDL cholesterol variable, seven studies provided data from 302 subjects which showed that the *Lupinus angustifolius* diet had a positive impact on LDL-C levels. The mean difference and corresponding 95% confidence interval were −0.17 (−0.27, −0.07) in favor of the experimental group, and the differences were considered significant (*p* = 0.001) (Figure 4).

### 3.4. Effects on TGs

A pooled data analysis from 302 subjects and seven studies showed that the *Lupinus angustifolius* diet had no impact on TG levels, although positive effects were observed between the intervention and control groups. The mean difference and corresponding 95% confidence interval were −0.06 (−0.23, 0.10) in favor of the experimental group, although the differences were not considered significant (*p* = 0.43) (Figure 5).

### 3.5. Effects on Body Mass Index

A pooled data analysis from 202 subjects and five studies showed that the *Lupinus angustifolius* diet had no impact on BMI, and no positive effects between the intervention and control groups were observed. The mean difference and corresponding 95% confidence interval were −0.33 (−1.50, 0.83) in favor of the experimental group, although the differences were not considered significant (*p* = 0.58) (Figure 6).

## 4. Discussion

The aim of this work was to investigate studies focusing on the biological properties of blue lupin (*Lupinus angustifolius*), its chemical components, and its importance for the control of biological and anthropometric health markers. The results of our study indicated that the blue lupin diet has a direct relationship with parameters such as blood glucose, weight, and LDL-C, but not with TGs or BMI.

### 4.1. Antidiabetic Properties

In general, our findings corresponded with several other studies that concluded that lupins are capable of favorably affecting blood pressure, blood lipids, insulin sensitivity, and intestinal microbiota [30]. Other studies reported that people who suffer from health conditions such as diabetes, hypertension, obesity, cardiovascular disease, hyperlipidemia, and colorectal cancer could benefit from the incorporation of this legume into their diets [4].

Due to their involvement on various levels, including the regulation of glucagon-like peptide 1 and their ability to block digestive enzymes, peptides produced from conglutin proteins in the blue lupin diet are essential for the regulation of glucose homeostasis [31]. In our study, we observed that the blue *lupin* diet had a direct relationship with parameters such as blood glucose. In this aspect, several studies reported on the effects of *lupin* proteins on blood glucose levels, where *lupin* proteins enriched with γ-conglutin reduced blood glucose levels, which was potentially due to the insulinotropic effects of this compound [20]. Additionally, γ-conglutin-derived peptides are absorbed through intestinal cell walls, enhancing their bioactive capacity in the regulation of glucose levels independently of the expression of SGLT1 and GLUT2 receptors [32,33] (Figure 7). It is possible that certain dipeptides and tripeptides, in combination with particular acidic amino acids, exhibit synergistic effects on glucose-stimulated insulin secretion. Additionally, protein hydrolysates were found to have a higher bioavailability than protein isolates. *Lupin* is a rich source of phytochemicals, most importantly bioactive peptides, alkaloids, polyphenols, phytosterols, tocopherols, etc. [19]. The significant concentrations of these phytochemicals and high fiber and protein contents help to control obesity and diabetes associated with the increase in body mass index, while on the other hand, the antioxidant, antihyperlipidemic, and anti-inflammatory activities of the phytochemicals of *lupin* act against various chronic diseases [20].

### 4.2. Effects on Weight and Body Mass Index

Overweight and obesity are considered to have the greatest impact on health in Western society [34]. The prevalence of obesity has increased exponentially, especially in industrialized countries, with diabetes, hypertension, and cardiovascular diseases as its most direct consequences [35,36,37]. Recently, the incorporation of legumes in hypocaloric diets has been demonstrated to reduce the levels of pro-inflammatory markers and improve several metabolic characteristics in overweight and obese individuals [38,39]. However, scientific evidence is scarce on their role in controlling parameters such as weight control, body mass index (BMI), or abdominal circumference. In this aspect, these studies coincide with our study, where we demonstrated no significant differences between weight control, BMI, and the *lupinus* diet.

The intake of foods rich in fiber, such as *blue lupin*, aids individuals in reaching a state of satiety more quickly that lasts longer over time [40]. In this sense, the high levels of resistant starch and dietary fiber present in legumes can have an effect on appetite control, increasing the feeling of satiety [41,42]. The regular consumption of pulses in diets reduces the risk of certain diseases. The consumption of legumes in a diet with a low content of saturated fatty acids aids in controlling lipid homeostasis and, as a consequence, decreases cardiovascular risks (the high fiber content of legumes, the low glycemic index, and the presence of phytochemicals are responsible for this property) [43]. In overweight and/or obese patients, the satiating effect of legumes aids in maintaining control over total intake, contributing to a negative energy balance [44,45,46].

### 4.3. Lipid-Lowering Effect

In our work, a pooled data analysis consisting of 302 subjects showed that the *Lupinus angustifolius* diet had a positive impact on LDL-C levels. In this aspect, we agree with several investigations focused on the lipid-lowering activity of different *lupin* constituents, such as fiber, proteins, and flour. Research has reported that proteins present in lupin reduce the expression of genes related to cholesterol synthesis [47]. The potential mechanism responsible for these health-promoting properties of lupin proteins is associated with increased low-density lipoprotein receptor activity [48]. Lupin proteins belong to the same plant family as soybeans; thus, their protein compositions are very similar. We assumed that the cholesterol-lowering effect of lupin proteins was due to their peptides or amino acids. It was suggested that conglutin c is one of the cholesterol-lowering peptides of lupin proteins [49], and bioactive peptides derived from soy protein hydrolysates have been reported to have cholesterol-lowering properties due to their stimulatory effects on LDL receptor transcription [50].

In contrast to the recently described TG-lowering effect of lupin proteins in hypercholesterolemic rats [51], this study found no favorable effects of lupin proteins on circulating TGs. In comparison, the effect of soy proteins on plasma TGs ranged from a decrease [52] to no effect [53]. Studies that focus on the effect of lupin proteins on plasma TGs should be conducted under controlled nutrient intake conditions since plasma TGs are influenced by several dietary factors, such as n-3 polyunsaturated fatty acids, refined carbohydrates, fatty acids, and alcohol [54].

### 4.4. Cardiovascular Protective Effect

According to the World Health Organization, cardiovascular disease (CVD) is the leading cause of death worldwide, accounting for 32% of all deaths in 2019 [55]. The magnitude of this problem is expected to increase in the coming years due to population aging and other causes of CVD, such as type 2 diabetes (DM2) and chronic kidney disease (CKD) [56].

The Di@betes study showed a 13.6% prevalence of DM2 in the adult population of Spain [57] and that CKD could affect up to 15.1% of the Spanish adult population in its different stages [56]. Deaths from both diseases are mainly due to CVDs [58,59]. These data represent high mortality rates and a significant consumption of resources by the National Health System.

In recent years, it has been recognized that this high cardiovascular risk rate is closely associated with the presence of vascular calcifications in these populations.

Furthermore, hyperphosphatemia and hyperglycemia present in patients with CKD and DM2, respectively, can stimulate calcification (Figure 8). In fact, in vitro assays with human vascular smooth muscle cells (VSMCs) demonstrated that both conditions led to a phenotypic transition of the cells, increasing the expression of markers typical of osteocytic cells, which is related to the calcification process [60]. This calcification is a consequence of tightly regulated mineralization processes that culminate in an organized deposition in the extracellular matrix by cells that have been transformed into an osteoblast-like phenotype [61], with processes similar to those found in normal osteogenesis [62]. These findings extend the link between bone remodeling and vascular calcification. Vascular calcification represents an important factor connecting cardiovascular events with bone loss, sharing pathophysiological mechanisms and genetic causes [63]. It has been shown that patients with DM2 and CKD present an increased risk of metabolic bone disease concomitant with CVDs. Therefore, there is growing interest in pharmacological agents that could inhibit bone loss and provide benefits in terms of slowing down the progression of CVDs, particularly in patients at high risk of developing CVDs, such as patients with DM2 and CKD. In this aspect, it has been shown that conglutin proteins from the leguminous plant blue lupin (*Lupinus angustifolius L.)* possess anti-inflammatory activity and decrease insulin resistance, which are properties that could be beneficial in the treatment of DM2 [18,63] and other pathological inflammatory processes, such as CVD and CKD [64].

### 4.5. Future Directions

The subject of lupin consumption, nutraceutical components, and health outcomes is a very new interesting area of investigation; therefore, more research is required to expand the evidence base. This should comprise multiple studies with similar aims, designs, and protocols based on adequately sized population groups, since scientific evidence reports a lack of scientific studies in humans.

Regarding health outcome measures, those for blood lipids, blood pressure glycemic control, and reductions in inflammation or cardioprotective properties would be the most useful in identifying the unique nutritional and physiological properties of lupin, the active components in the lupin protein, and the minimum dose required for beneficial health effects.

## 5. Conclusions and Perspectives

*Lupinus angustifolius* has been widely studied as an antidiabetic agent based on the effects that its different components can exert on different levels of human physiology. *Lupinus* protein hydrolysates have been shown to improve glucose homeostasis in patients with type 2 diabetes mellitus (DM2) or those who are glucose intolerant, as they increase insulin secretion. In comparison to other protein types, lupin proteins have the potential to improve the LDL-C ratio of hypercholesterolemic subjects; however, the same does not occur with TGs, which require a more controlled dietary environment. In this aspect, further research would be necessary to elucidate the active components of lupin proteins and the minimum dose required for beneficial health effects. As we have observed, no scientific evidence has been reported on the proteins’ effects on weight or BMI, which we believe would depend on the timing of the diet, supplementation, and exercise. Regarding cardioprotective activity, the phenolic content and composition of *L. angustifolius* may have positive implications for reducing the risk of cardiovascular disease due to its protective effect on blood vessel health. The research described in this review clearly indicates that *L. angustifolius* may play an important role in the dietary prevention of hyperlipidemia and hypertension. Therefore, it would be highly advisable to increase its consumption in diets. However, further studies, ideally in humans, are required to truly establish *L. angustifolius’s* health-promoting properties.

## Figures and Tables

**Figure 1 foods-12-02749-f001:**
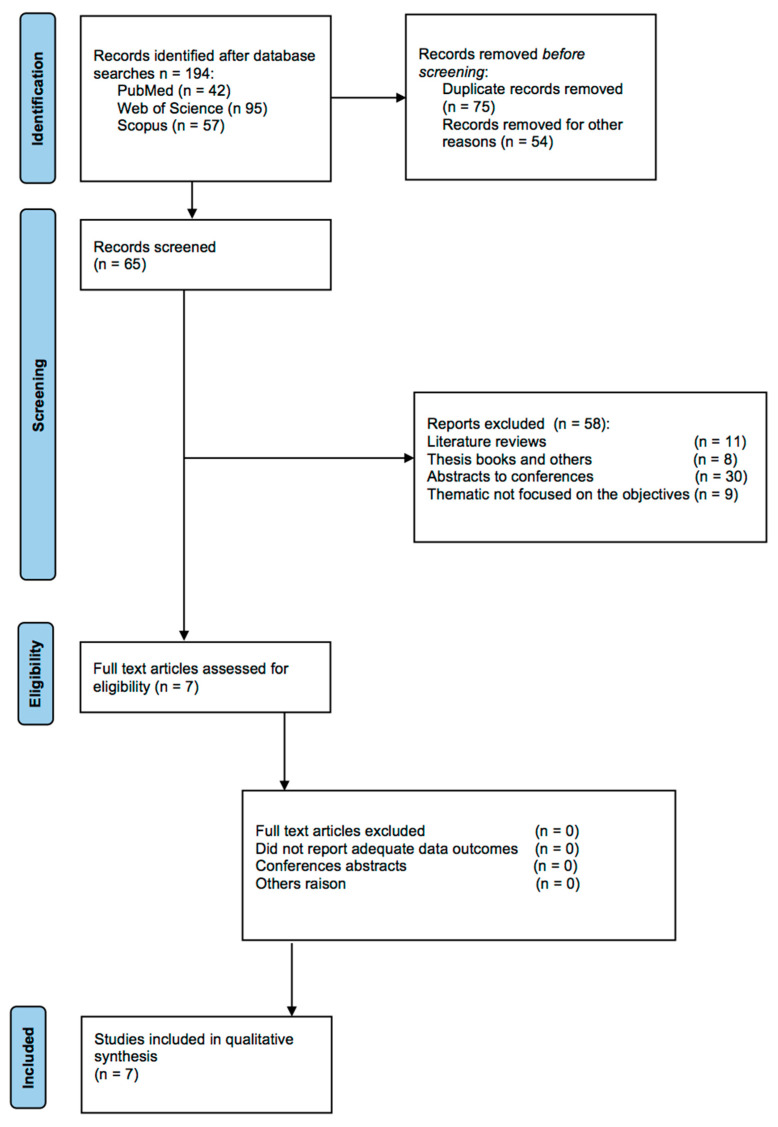
Schematic representation of the search strategy according to PRISMA guidelines.

**Figure 2 foods-12-02749-f002:**
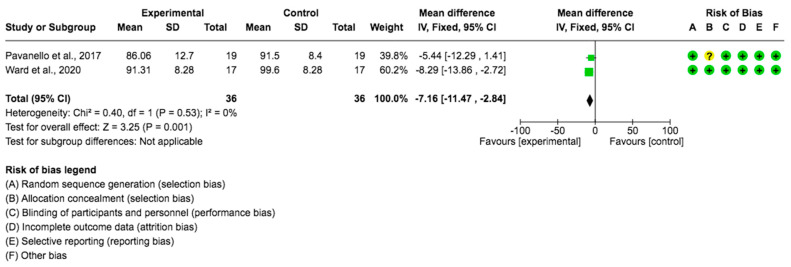
Forest plot for glycemia [28,29].

**Figure 3 foods-12-02749-f003:**
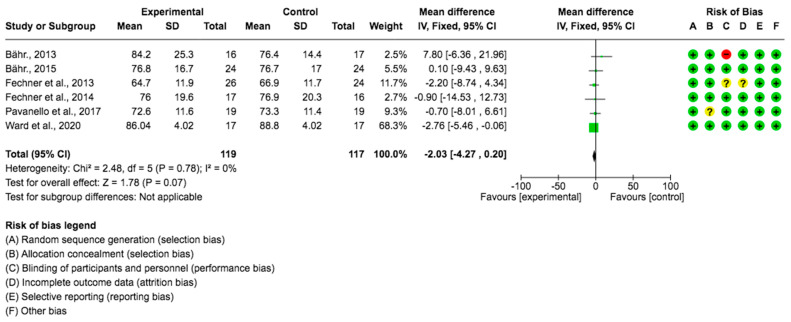
Forest plot for weight [23,24,26,27,28,29].

**Figure 4 foods-12-02749-f004:**
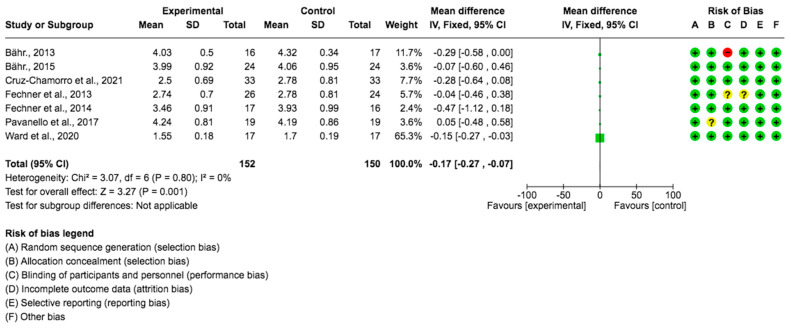
Forest plot for LDL-C [23,24,25,26,27,28,29].

**Figure 5 foods-12-02749-f005:**
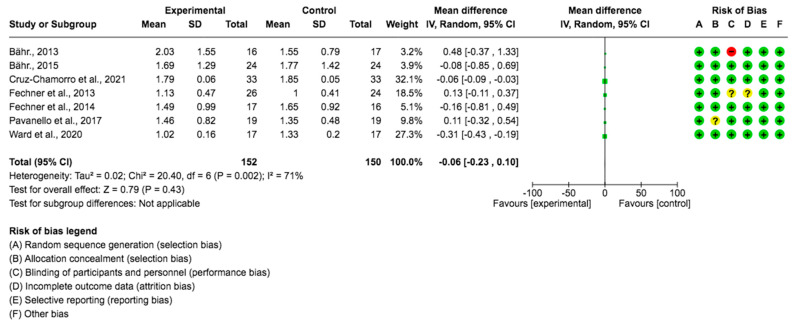
Forest plot for TGs [23,24,25,26,27,28,29].

**Figure 6 foods-12-02749-f006:**
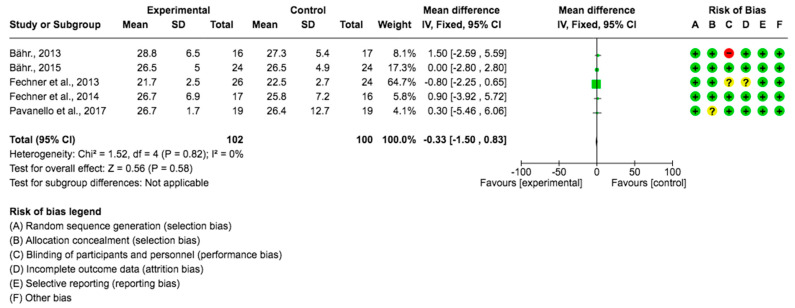
Forest plot for body mass index [23,24,26,27,28].

**Figure 7 foods-12-02749-f007:**
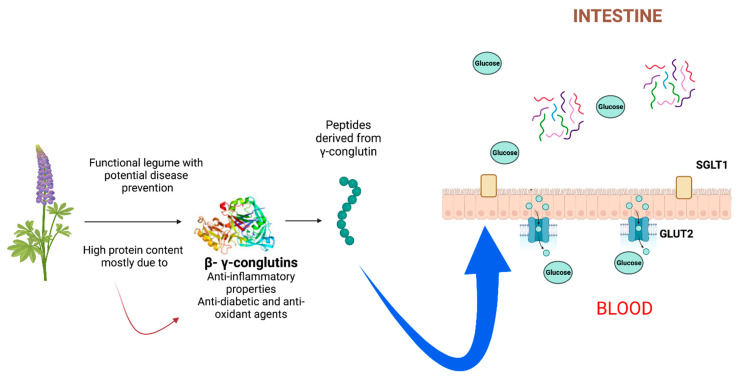
Conglutin proteins, from the leguminous plant blue lupin (*Lupinus angustifolius L*.), possess anti-inflammatory and antidiabetic activities and could be useful in inflammatory processes. Peptides derived from γ-conglutin could be absorbed through the intestinal cell wall being responsible for their bioactive capacity in the regulation of glucose levels. γ-Conglutines are also capable of promoting a decrease in glucose uptake both in cells and in the jejunum independently of the expression of the SGLT1 and GLUT2 transporters.

**Figure 8 foods-12-02749-f008:**
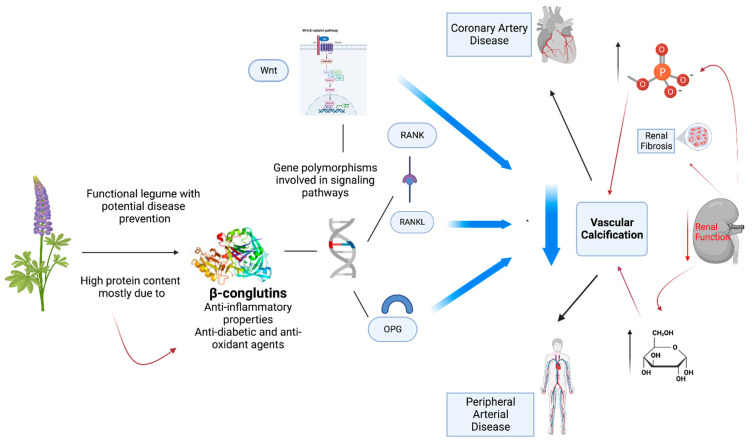
The decrease in renal mass causes a decrease calcitriol and an increase in phosphorus (P). The increase in P leads to a greater decrease in calcitriol and an increase in fibroblast growth factor 23 (FGF23) and parathyroid hormone (PTH). PTH also increases in an attempt to palliate the calcitriol deficit and increase FGF23. This network increases cardiovascular risk: ventricular hypertrophy, atherosclerosis, endothelial dysfunction, and vascular calcifications.

**Table 1 foods-12-02749-t001:** Characteristics and summary of the studies included in the review.

Reference	Study Type	Subjects (*n*) and Characteristics	Intervention	Control/ Comparator	Main Health Markers	Main Outcomes	Country
Bähr et al., 2013 [23]	RCT double-blind cross-over study; 8-week treatment, 4-week washout	*n* = 33 hypercholesterolemic (TC ≥ 5.2 mmol/L) men (*n* = 33) and women (*n* = 18) Mean age range 49.4 ± 13.9–49.7 ± 12.8 years Mean BMI range 27.3 ± 5.4–28.8 ± 6.5 kg/m^2^	Blue lupin protein isolate (LPI) protein drinks, 25 g LPI per day	Milk protein isolate (MPI) protein drinks, 25 g MPI per day	TC, LDL, HDL, LDL:HDL, TGs, 4- and 8-week BW, SBP, DBP, resting pulse, urea, and hs-CR	↑ HDL at week 4 for LPI compared to MPI (*p* = 0.036) No difference between treatments for lipids ↓ LDL for both treatments at 4 weeks but not at 8 weeks (*p* ≤ 0.008) ↓ LDL:HDL for LPI (*p* = 0.022) Both treatments slight ↑ BW and body fat from 0–8 weeks (*p* ≤ 0.045) No difference between treatments	Germany
Bähr et al., 2015 [24]	RCT double-blind cross-over 3-phase study; 28-day treatment, 6-week washout	*n* = 72 *n* = 68 completed, hypercholesterolemic (TC ≥ 5.2 mmol/L) men (*n* = 28) and women (*n* = 40) Mean age range 50.4 ± 19.2–59.8. ± 9.3 years Mean BMI range 24.9 ± 5.0–27.6 ± 4.4 kg/m^2^	Blue lupin protein isolate, 25 g consumed daily in 4 food products	Milk protein (MP) 25 g in 4 food products; MP foods plus 2.5 g/d arginine in capsule form (MPA); placebo capsules added to LP and MP diets for blindness	TC, LDL, HDL, LDL:HDL, oxidized LDL, TGs, SBP, DBP hs-CRP, urea, uric acid, and homocysteine	↓ LDL after lupin compared with MP (*p* = 0.044) ↓ 0–28 d TC (*p* < 0.001), LDL (*p* < 0.01), and HDL (*p* < 0.001) after lupin and MPA ↓ TGs (*p* < 0.05) after lupin ↑ BW and body fat	Germany
Chamorro et al., 2005 [25]	RCT single-blind cross-over study; 28 days of treatment	*n* = 35 *n* = 33 completed, healthy subjects Mean age 30.27 ± 1.72 Mean BMI 22.71 ± 0.43 kg/m^2^	Ingest an LPHb containing 1 g LPHs for 28 days	No significant differences (*p* > 0.05) in the age and body mass index (BMI) between males and females were observed at baseline; after 28 days of intake no impact in the weight of volunteers was observed (day 0: 65.81 ± 1.94 kg; day 28: 66.16 ± 1.95 kg; *p* = 0.082)	TC, LDL, HDL, LDL:HDL, LDL-C/HDL-C, TGs, SBP, and DBP	↓TC, LDL, TC:HDL, and LDL:HDL for both treatments (*p* < 0.05) ↓ TC (*p* = 0.001), LDL (*p* = 0.001), TC:LDL (*p* = 0.001), and LDL:HDL (*p* = 0.003) for lupin relative to control	Spain
Fechner et al., 2013 [26]	RCT double-blind cross-over study; 4 periods of 2 weeks each: run-in, 2 treatments, and washout	*n* = 76 healthy men (*n* = 21) and women (*n* = 55). Mean age 24.4 ± 3.2 years Mean BMI 21.7 ± 2.4 kg/m^2^	Blue lupin kernel fiber and white lupin kernel fiber Total dietary fiber per treatment 25 g/d in beverages	Citrus fiber as active comparator for 2 lupin and 1 soya fiber treatment	TC, HDL, LDL, and TGs	No significant changes in serum lipids for all treatments; propionate and n-butyrate excretion for blue lupin (*p* ≤ 0.05). citrus; blue and white lupin from run-in (*p* ≤ 0.05); ↓ BW and body fat	Germany
Fechner et al., 2014 [27]	RCT double-blind cross-over study; 3 intervention periods of 4 weeks each, run-in and 2 washout periods of 2 weeks each	*n* = 52 moderately hypercholesterolemic (TC > 5.2 mmol/L) men (*n* = 20) and women (*n* = 32) Mean age: 46.9 ± 3.2 years. Mean BMI: 26.5 ± 5.9 kg/m^2^	Blue lupin kernel fiber 25 g/d	Citrus fiber 25 g/d as active comparator; control diet (CD) with no added fiber	General excretion markers, fecal concentration, or excretion of neutral sterols, bile acids, and SCFAs BW, body composition, BP, TC, HDL, LDL, TGs, LDL:HDL, hs-CRP, and satiety score	↓ BW, BMI, and WC from baseline (*p* ≤ 0.001) and against control (*p* ≤ 0.01)↓ TC (9%), LDL (12%), and TGs (10%) for lupin compared with citrus (*p* ≤ 0.02) ↓ SBP (*p* = 0.01) for lupin compared to baseline ↑ Perception of satiety (*p* ≤ 0.001)	Germany
Pavanello et al., 2020 [28]	Randomized parallel-group double-blind single-center study; 12-week intervention	*n* = 25 healthy subjects Mean age 55.3 ± 54.66 years Mean BMI 26.66 ± 4.5 kg/m^2^	Half of the participants consumed a lupin protein concentrate (30 g/day of protein), the other half a lactose-free skimmed milk powder (30 g/day of protein), both integrated into a mixed low-lipid diet	Received all bags necessary for one-month treatment during their visits to the clinical center and were instructed to add the powders to their normal foods, avoiding extensive cooking	TC, LDL, HDL, LDL:HDL, oxidized LDL, TGs, SBP, DBP hs-CRP, and PCSK9	↓ TC (*p* < 0.001) ↓ LDL (9%), LDL (8%), HDL (7.5%), and TGs (10%) ↓ FG Significant changes in body weight (−1.7%, *p* < 0.05) and BMI (1.5%, *p* < 0.023) were found with the milk diet, whereas a decrease in WC/HC was recorded only with the lupin diet (2%, *p* < 0.047)	Italy
Ward et al., 2020 [29]	RCT double-blind cross-over study; 1-week run-in period, 2 × 8-week treatment with 8-week washout period	*n* = 22 *n* = 17 completed, men (*n* = 14) and women (*n* = 8) with moderate- to well-controlled type 2 diabetes (HbA1c < 9%) Mean age 58 ± 6.6 years Mean BMI 29.9 ± 3.5 kg/m^2^	Lupin-enriched foods replacing 20% of daily energy intake; consumed every breakfast and lunch and at least 3 dinners per week; average daily intake ~45 g lupin per day (12 g/d protein 10 g/d fiber)	Wheat-based control foods	FG (at waking, 1 h post breakfast, immediately pre-lunch, and 1 h post-lunch), HOMA-IR, BW, BP, TC, LDL, TGs, HDL, and C-peptide	No difference between treatments Borderline significant decrease in TGs with lupin ↓ FG, ↓ body weight, and ↓ LDL	Australia

Abbreviations: body mass index (BMI); body weight (BW); diastolic blood pressure (DBP); high-density lipoprotein cholesterol (HDL); homeostasis model assessment of insulin resistance (HOMA-IR); high-sensitivity C-reactive protein (hs-CRP); interleukin-6 (IL-6); low-density lipoprotein cholesterol (LDL); plasma glucose (PG); systolic blood pressure (SBP); serum glucose (SG); serum insulin (SI); soluble intracellular cell adhesion molecule-1 (sICAM-1); triglycerides (TGs); total cholesterol (TC); proprotein convertase subtilis/kexin type 9 (PCSK9); homeostatic model assessment of insulin resistance (HOMA-IR).

## Data Availability

The data used to support the findings of this study can be made available by the corresponding author upon request.

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
