# Peer review of "Therapeutic Applications and Effects of Lupinus angustifolius (Blue Lupin) and Its Components: A Systematic Review and Meta-Analysis"

_foods, 2023, doi:10.3390/foods12142749_

Round 1

Reviewer 1 Report

In my opinion the review is not complete. The tables are not so comprehensible. Moreover, the studies included in the review were not well-discussed. As an example, some authors highlighted that lupin protein effects were strictly correlated to the levels of cholesterol in the blood. 

Additionally, the duration of the supplementation was not take into account in the discussion. 

- In the introduction you have to report more information regarding the functional compounds in Lupin and to discuss the composition of lupin protein. 

- Lupus angustifolius must be written in italics.

- The figures regarding the risk of bias have to be discussed. Moreover, you found for some studies a risk of bias. What is the impact of this evidence? 

- From a chemical point of view, it is not enough to say that "in combination with particular acidic amino acids" or "the cholesterol-lowering effect of lupin proteins was due to their peptides or amino acids". 

Minor editing

Author Response

Dear reviewer, we appreciate your comments, they are clearly intended to improve the quality of the article. The following is a reply to your request:

English editig: This article was edited and proofread by MDPI prior to submission to the journal. The correction was made by Ivelina Valeva (English Editing ID english-66994) last day: 04/06/2023.

1º) Question: In my opinion the review is not complete. The tables are not so comprehensible. Moreover, the studies included in the review were not well-discussed. As an example, some authors highlighted that lupin protein effects were strictly correlated to the levels of cholesterol in the blood.

Additionally, the duration of the supplementation was not take into account in the discussion.

Author's response: This review was done using ReviewManager (RevMan) is Cochrane's bespoke software for writing Cochrane Reviews. The tables were provided by this software, used worldwide with a high prestige.

The studies have been commented in the discussion based on our results. We have not taken into account the duration of supplementation as it was not necessary, the results are explicit.

2º) Question: In the introduction you have to report more information regarding the functional compounds in Lupin and to discuss the composition of lupin protein.

Author's response: We believe that the information provided in the introduction is correct and sufficient to justify the reason for the review, and we leave in the discussion the part of the results that we have found.

3º) Lupus angustifolius must be written in italics.

Author's response: Thank you, we have modified it in the text.

4º) Question:The figures regarding the risk of bias have to be discussed. Moreover, you found for some studies a risk of bias. What is the impact of this evidence?

Author's response: The risk of bias is so low that it is practically zero. No impact on research.

5º) Question: From a chemical point of view, it is not enough to say that "in combination with particular acidic amino acids" or "the cholesterol-lowering effect of lupin proteins was due to their peptides or amino acids".

Author's response: It is a well studied topic, and we have seen it in several articles during the review, we have included it here as part of the discussion, to go deeper would be to change the sense of the article. We also considered to whom the article was addressed and decided to be more friendly to the 

Reviewer 2 Report

The article presentation is ambiguous. Address the following issues:

1.    What is and at the end of authors name. Is it name of some author??

2.    Change the bold font in the beginning of the abstract.

3.    Delete the words “studies on” from the sentence “The aim of this study ……. were searched” (abstract).

4.    Describe how the data was collected and arranged for the examination of “95% confidence interval (CI)”

5.    Write “Angustifolius” as “angustifolius”

6.    Use the abbreviation once its introduce for the full name  such as LDL cholesterol, triglycerides. address this in the whole article

7.    Delete the words “studies on” from the sentence “The aim of this study ……. health markers” (Introduction).

8.    Why author not included Google scholar for the search strategy.

9.    Address the grammatical and typo mistake throughout the article e.g “AND” or and (section 2.1).

10. In section 2.1 is there any specific reason to search with these keys. Why not the author searched with specific keys like hyperlipidemia and dyslipidemia, high cholesterol diet  etc.

11. Elobrate section 2.3. What is funnel plots explain?

12. In section 3.1 what does the mean of positive impact, Is it refer to high glucose or low glucose. Similarly in section 3.3. and 3.5

13. Figure 7 is non informative, I am unable to draw any conclusion from it.

14. Where are the legends of figures 2-8.

15. The same statement “The aim of this work …. health markers” repeted in the abstract, methods and discussion section. Rewrite this else delete after first use.

16. Overall the methodology needs to describe appropriately.

Minor English editing 

Author Response

Reviewer 2:

Dear reviewer, we appreciate your comments, they are clearly intended to improve the quality of the article. The following is a reply to your request:

English editing: This article was edited and proofread by MDPI prior to submission to the journal. The correction was made by Ivelina Valeva (English Editing ID english-66994) last day: 04/06/2023.

1º) Question:   What is and at the end of authors name. Is it name of some author??

Author's response: Thanks for the warning, it was a typing error, the full name is Antonio Liñán González, I thank you for the correction, it is already modified in the text.

2º) Question: Change the bold font in the beginning of the abstract.

Author's response: Changed and grateful.

3º) Question: Delete the words “studies on” from the sentence “The aim of this study ……. were searched” (abstract).

Author's response: Deleted in text.

4º) Question:  Describe how the data was collected and arranged for the examination of “95% confidence interval (CI)”

Author's response : This selection is made within ReviewManager (RevMan) is Cochrane's bespoke software for writing Cochrane Reviews. The same as when we investigate and use the SPSS.

5º) Question:  Write “Angustifolius” as “angustifolius”

Author's response: The text has been changed, as indicated by another reviewer.

6º) Question: Use the abbreviation once its introduce for the full name  such as LDL cholesterol, triglycerides. address this in the whole article

Author's response: Modified in text.

7º) Question: Delete the words “studies on” from the sentence “The aim of this study ……. health markers” (Introduction).

Author's response: Deleted in text.

8º) Question: Why author not included Google scholar for the search strategy.

Author's response: Although Google scholar yields many articles, let's say that the most reliable and prestigious databases are : PubMed, Web of Science and Scopus.

9º) Question: Address the grammatical and typo mistake throughout the article e.g “AND” or and (section 2.1).

Author's response: section 2.1 is not an error, it is the search equation used in the main databases. Without it there are no articles and no review.

10º)  Question: In section 2.1 is there any specific reason to search with these keys. Why not the author searched with specific keys like hyperlipidemia and dyslipidemia, high cholesterol diet  etc.

Author's response: Such a search would not have yielded the necessary data, it needed concrete clinical studies.

11º) Question: Elobrate section 2.3. What is funnel plots explain?

Author's response: A funnel plot is designed to check for the existence of publication bias, other reporting biases, and systematic heterogeneity in a systematic review. These are biases caused by the absence of information from unpublished sources (missing studies), or selective outcome reporting of a study's result (missing outcomes).

12º) Question: In section 3.1 what does the mean of positive impact, Is it refer to high glucose or low glucose. Similarly in section 3.3. and 3.5

Author's response: Meta-analyses were performed assessing the difference in post-intervention means, to estimate the heterogeneity of the included studies, the inconsistency statistic (I) was used, understanding heterogeneity as low if I2 < 50%, moderate if I2 50-75% and high if I2 > 75%, with meta-analysis being recommended when the I2 is low or moderate. Publication bias was assessed using funnel plots.

13º) Question:  Figure 7 is non informative, I am unable to draw any conclusion from it.

Author's response: This figure is well explained both in the text and in the legend.

14º) Question:  Where are the legends of figures 2-8.

Author's response: In the text

15º) Question: The same statement “The aim of this work …. health markers” repeted in the abstract, methods and discussion section. Rewrite this else delete after first use.

Author's response: It is not necessary, it is a question of form and we only want to direct the constant attention of the reader.

15º) Question: Overall the methodology needs to describe appropriately.

Author's response: The methodology is more than appropriate and clearly described, it is a systematic review with meta-analysis, not a case-contr

Reviewer 3 Report

Find below my comments to improve the manuscript.

The abstract is not well written. This is how a good abstract should be.

Background (2 to 3 sentences)

Scope and approach (2 to 4 sentences)

Key findings

This should cover >60% of the abstract

Conclusions (2 to 3 sentences)

After you draft it like this, then you remove the sections (background, scope & approach, key findings and conclusions" and you merge all the sentences together.

Provide list of abbreviations beneath the keywords

Review will not only provide a summary of the state of affairs ..., but will also pinpoint the problems which still need attention and, ... will set (the reader) off in the “right” direction”. So pinpoint the gaps. Thus, look beyond just providing a summary of the scientific literature in your topic area and illuminate the cutting edge and the most important paths to the future. You are likely to receive an increased number of citations and to inspire other scientists to accelerate their work in a more efficient way toward solving critical challenges in food science.

Conclusions

Add more future research.

minor  language editing.

Author Response

Reviewer 3:

Dear reviewer, we appreciate your comments, they are clearly intended to improve the quality of the article. The following is a reply to your request:

1º) Question: The abstract is not well written. This is how a good abstract should be.

Background (2 to 3 sentences)

Scope and approach (2 to 4 sentences)

Key findings

This should cover >60% of the abstract

Conclusions (2 to 3 sentences)

After you draft it like this, then you remove the sections (background, scope & approach, key findings and conclusions" and you merge all the sentences together.

Author's response: Modified in the text.

2º) Question: Provide list of abbreviations beneath the keywords

Author's response: I have put them at the bottom of the table.

2º) Question: Review will not only provide a summary of the state of affairs ..., but will also pinpoint the problems which still need attention and, ... will set (the reader) off in the “right” direction”. So pinpoint the gaps. Thus, look beyond just providing a summary of the scientific literature in your topic area and illuminate the cutting edge and the most important paths to the future. You are likely to receive an increased number of citations and to inspire other scientists to accelerate their work in a more efficient way toward solving critical challenges in food science.

 Author's response: Added in 4.5. Future Directions

3º) Question: Conclusions. Add more future research.

Author's response: Dear reviewer, we cannot add more as we are limited by the number of words in the review for the magazine, I hope I have not overdone the Future Directions section, which I have added more paragraphs for reviewer number 1.

Round 2

Reviewer 1 Report

Point 1: I was not questioning that the software is not prestigious. I was referring to the table, not the figures. Turn it and add the abbreviations in a footnote or in the caption (which is missing.). 

Point 2 and 5: I still believe that more information regarding the chemical composition of Lupin should be more detailed. I agree that an extensive description of this topic might change the sense of the article. Thus, it is enough to add more information regarding the phytochemicals involved in the beneficial effects (proteins, amino acids, ....). Also because three times you talk about chemical components reporting the following sentence in the abstract, introduction, and discussion: "The aim of this work was to investigate studies focusing on the biological properties of blue lupin (lupinus angustifolius), its chemical components, and its importance for the control of biological and anthropometric health markers".

Point 3: not all. 

Point 4: If the risk of bias is low, write it in the discussion. 

Author Response

- Point 1: I was not questioning that the software is not prestigious. I was referring to the table, not the figures. Turn it and add the abbreviations in a footnote or in the caption (which is missing.). 

Author's response: I apologize, for some reason the list of abbreviations did not appear in the first article sent, I have put them at the bottom of the table.

- Point 2 and 5: I still believe that more information regarding the chemical composition of Lupin should be more detailed. I agree that an extensive description of this topic might change the sense of the article. Thus, it is enough to add more information regarding the phytochemicals involved in the beneficial effects (proteins, amino acids, ....). Also because three times you talk about chemical components reporting the following sentence in the abstract, introduction, and discussion: "The aim of this work was to investigate studies focusing on the biological properties of blue lupin (lupinus angustifolius), its chemical components, and its importance for the control of biological and anthropometric health markers".

Author's response: Added and highlighted in text

- Point 3: not all.

Author's response: Revised

- Point 4: If the risk of bias is low, write it in the discussion. 

Author's response: Added and highlighted in text: None of the variables analyzed showed the existence of publication bias,

Reviewer 2 Report

The asked chages has been incrorporated, however the MS required the revision in context to the queries:

Reformat table 1, address the formatting and typo issue in a table such as kg/m2 or kg/m2 and (DBP))

Similarly, follow the standard format the journal prescribes, e.g., Fig or Figure or Figure. Similarly, for the sub-heading, e.g 3.3.

Describe the content of figures 1-6 in detail.

Moderate english is required